# Long-Lasting Exendin-4 Fusion Protein Improves Memory Deficits in High-Fat Diet/Streptozotocin-Induced Diabetic Mice

**DOI:** 10.3390/pharmaceutics12020159

**Published:** 2020-02-16

**Authors:** Kyung-Ah Park, Zhen Jin, Jong Youl Lee, Hyeong Seok An, Eun Bee Choi, Kyung Eun Kim, Hyun Joo Shin, Eun Ae Jeong, Kyoung Ah Min, Meong Cheol Shin, Gu Seob Roh

**Affiliations:** 1Department of Anatomy and Convergence Medical Science, Bio Anti-Aging Medical Research Center, Institute of Health Sciences, College of Medicine, Gyeongsang National University, Jinju, Gyeongnam 52727, Korea; imkapark@gmail.com (K.-A.P.); zkim777@gmail.com (Z.J.); jyv7874v@naver.com (J.Y.L.); gudtjr5287@hanmail.net (H.S.A.); dmsql2274@naver.com (E.B.C.); kke-jws@hanmail.net (K.E.K.); k4900@hanmail.net (H.J.S.); jeasky44@naver.com (E.A.J.); 2College of Pharmacy and Inje Institute of Pharmaceutical Sciences and Research, Inje University, 197 Injero, Gimhae, Gyeongnam 50834, Korea; minkahh@inje.ac.kr; 3College of Pharmacy and Research Institute of Pharmaceutical Sciences, Gyeongsang National University, Jinju, Gyeongnam 52828, Korea

**Keywords:** exendin-4, fusion protein, Glucagon-like peptide 1R (GLP-1R), memory deficits, hippocampus, diabetic mice

## Abstract

Glucagon-like peptide 1 (GLP-1) mimetics have been approved as an adjunct therapy for glycemic control in type 2 diabetic patients for the increased insulin secretion under hyperglycemic conditions. Recently, it is reported that such agents elicit neuroprotective effects against diabetes-associated cognitive decline. However, there is an issue of poor compliance by multiple daily subcutaneous injections for sufficient glycemic control due to their short duration, and neuroprotective actions were not fully studied, yet. In this study, using the prepared exendin-4 fusion protein agent, we investigated the pharmacokinetic profile and the role of this GLP-1 mimetics on memory deficits in a high-fat diet (HFD)/streptozotocin (STZ) mouse model of type 2 diabetic mellitus. After induction of diabetes, mice were administered weekly by intraperitoneal injection of GLP-1 mimetics for 6 weeks. This treatment reversed HFD/STZ-induced metabolic symptoms of increased body weight, hyperglycemia, and hepatic steatosis. Furthermore, the impaired cognitive performance of diabetic mice was significantly reversed by GLP-1 mimetics. GLP-1 mimetic treatment also reversed decreases in GLP-1/GLP-1 receptor expression levels in both the pancreas and hippocampus of diabetic mice; increases in hippocampal inflammation, mitochondrial fission, and calcium-binding protein levels were also reversed. These findings suggest that GLP-1 mimetics are promising agents for both diabetes and neurodegenerative diseases that are associated with increased GLP-1 expression in the brain.

## 1. Introduction

Type 2 diabetes mellitus (T2DM) is a widespread, chronic inflammatory disease associated with long-term complications including insulin resistance and cardiovascular diseases. Numerous studies have suggested a close causal association between T2DM and cognitive impairment [1,2]. Furthermore, T2DM has been implicated as a risk factor in not only vascular dementia but also in Alzheimer’s disease (AD) [3]. Insulin resistance in peripheral tissues influences insulin resistance in the brain thereby affecting cognition [4]. Several antidiabetic drugs exert beneficial effects on both systemic and neuronal metabolic processes, which could be clinically significant for the treatment of diabetes-induced complications such as cognitive decline. However, diabetes is caused by multiple mechanisms and its etiology is not fully understood.

Glucagon-like peptide-1 (GLP-1) is an insulinotropic hormone secreted by the intestines that decreases appetite, elicits neuroprotective activities, and promotes insulin secretion to reduce blood glucose levels [5,6,7]. However, the development of long-lasting GLP-1 receptor (GLP-1R) agonists, such as exendin-4 and liraglutide, and various dipeptidyl peptidase-4 (DDP-4) inhibitors have helped reveal the diverse functions of GLP-1R [8,9]. Studies have demonstrated the anti-obesity, antidiabetic, and anti-inflammatory effects of exendin-4 and its effectiveness in improving cognition, in both AD and diabetic patients [10,11,12,13]. Improved brain function is thought to result from secondary GLP-1R antidiabetic and anti-inflammatory effects in the brain [14]. Nevertheless, it is unclear how neuronal activity is affected by GLP-1 mimetics.

This study introduces the development and in vivo characterization of a novel long-lasting GLP-1 mimetic exendin-4 fusion protein (ExA) composed of exendin-4 genetically fused to an albumin binding domain (ABD) and an anti-FcRn affibody. The ABD allows the ExA to instantly bind to serum albumins, once it enters the blood circulation. In addition, the high-affinity binding of ExA to the endothelial FcRn mediated by both albumin and the anti-FcRn affibody would allow efficient FcRn-mediated recycling, eventually leading to a great increase of plasma half-life [15]. After the preparation of the ExA and characterization of the pharmacokinetic (PK) profiles, its effects on body weight, blood glucose levels, and cognitive function were assessed in high-fat diet (HFD)/streptozotocin (STZ)-induced diabetic (HFD/STZ) mice. Moreover, the effects of ExA on appetite suppression, insulin resistance, and hepatic steatosis, as well as its protective effects on neuroinflammation, mitochondrial fission, and calcium regulation in the hippocampus, were studied in these mice.

## 2. Materials and Methods

### 2.1. Materials

T4 DNA ligase, NdeI, and EcoRI were obtained from New England Biolabs (Ipswich, MA, USA). Alexa Fluor 488- and 594-conjugated donkey secondary antibodies were obtained from Invitrogen (Carlsbad, CA, USA). Streptozotocin (STZ), D-glucose, and some reagents for the western blot analysis such as mouse anti-glial fibrillary acidic protein (GFAP), β-actin, and α-tubulin were purchased from Sigma-Aldrich Co. (St. Louis, MO, USA). Primary antibodies such as GLP-1, Tau, NF-κBp65, insulin receptor (IR)-β, and calbindin were purchased from Santa Cruz Biotechnology, Inc (Santa Cruz, CA, USA). Rabbit anti-parvalbumin, GLP-1R, hippocalcin, nuclear p84, and mitochondrial VDAC1 were obtained from Abcam (Cambridge, MA, USA). Isopropyl-β-thiogalactopyranoside (IPTG), LB broth, p-Tau, and 4′,6-diamidino-2-phenylindol (DAPI) were obtained from the Thermo Fisher Scientific (Waltham, MA, USA). Total dynamin related protein1 (Drp1) and optic atrophy1 (OPA) were obtained from BD Bioscience (Franklin Lakes, NJ, USA).

### 2.2. Expression and Purification of ExA

The pET-ExA expression vector was prepared by double digestion of the genes encoding exendin-4-ABD-anti-FcRn affibody (570 bp) (Genscript Biotech, Piscataway, NJ, US) using restriction enzymes (NdeI & EcoRI) and subsequent ligation to pET28a-SUMO vector. The scheme of the pET-ExA is depicted in Figure 1A. The *Escheridhia coli* expression system was used for producing the ExA fusion protein. First, a colony of *E. coli* transformed with pET-ExA was added into 40 mL of fresh LB medium containing 80 μg/mL of kanamycin, and this culture was incubated overnight at 37 °C under the shaken condition at 220 rpm. Next, the culture was diluted with 1 L of LB medium (80 μg/mL of kanamycin) and incubated under the identical conditions for the starter culture. When the OD_600_ reached 1, IPTG (final 0.5 mM) was added as the inducer for the production of ExA. The culture was further maintained under the same condition for 4 h. After incubation, the *E. coli* cells were collected by centrifugation and then re-dispersed in the lysis buffer (20 mM PBS with 300 mM NaCl, pH 7). The *E. coli* cells were lysed by sonication and, after centrifugation, the supernatant fraction containing the soluble ExA proteins was applied to Talon resins (TALON Superflow resin, GE Healthcare Bio-Sciences, Pittsburgh, PA, US) for purification. Briefly, after loading the cell lysates to the Talon resins, the resins were washed with 20 mM PBS containing 300 mM NaCl (pH 7) and then the ExA was eluted by adding the eluent buffer (20 mM PBS; 300 mM NaCl, and 300 mM imidazole). The production and purification of ExA was monitored by SDS-PAGE analysis. The final ExA product was kept in the refrigerator at 4 °C until further use.

### 2.3. Animal Care

Male ICR (6 weeks of age) and C57BL/6 mice (3 weeks of age) were purchased (KOATECH (Pyeongtaek, South Korea) and housed in the animal facility at Gyeongsang National University (GNU). Institutional Board of Research at GNU (GNU-160530-M0025) approved all animal experiments (24 October 2016). We performed in accordance with the National Institutes of Health guidelines for laboratory animal care. Mice were kept in a light/dark cycle of 12 h.

### 2.4. Pharmacokinetic (PK) Studies

Male ICR mice were subcutaneously (s.c.) administered with 1.6 mg/kg of ExA (*n* = 3). A blood specimen was withdrawn from each mouse at 0, 10 min, 1 h, 3 h, 8 h, 1 day, 2 day, 3 day, 4 day, 7 day, 10 day, and 14 day post-administration. The blood was collected from the venous sinus using a capillary tube, and the plasma sample was acquired by centrifugation of the blood at 1000 rpm for 5 min. The plasma samples were analyzed by using the exendin-4 enzyme immunoassay kit (EK-070-94, Phoenix Pharmaceuticals, Inc., Burlingame, CA, USA), according to the manufacturer’s instruction. Based on the plasma concentration-versus-time profiles of the ExA, the pharmacokinetic profiles of the plasma samples were analyzed by using non-compartmental analysis (NCA) with Phoenix^®^ WinNonlin^®^ software (Certara LP. Princeton, NJ, US). The PK parameters such as the half-life (*t*_1/2_), the maximum plasma concentration (*C*_max_), and the time to reach the *C*_max_ (*T*_max_) were calculated.

### 2.5. Induction of T2DM and Experimental Design

T2DM was induced in mice by feeding animals a HFD (60% kcal fat; Research Diet, New Brunswick, NJ, USA) in combination with STZ treatment (100 mg/kg, Sigma-Aldrich, St. Louis, MO, USA), as previously described [16]. Briefly, mice (*n* = 20) were fed a HFD for 16 weeks, STZ was administered via intraperitoneal injection, and a HFD feeding was continued for an additional 4 weeks. After diabetic induction, mice (*n* = 10) were administered ExA (6.4 mg/kg) via weekly intraperitoneal injection for 6 weeks. ExA dose and treatment duration were selected based on preliminary experiments (data not shown). HFD/STZ mice (*n* = 10) were administered 0.9% sterile saline vehicle as a control. Normal diet (ND)-fed mice (*n* = 3–8 per group) were administered either vehicle or ExA (6.4 mg/kg). Following treatment, glucose tolerance test (GTT) and Morris Water Maze (MWM) test were performed (Figure 1A). Body weight and food intake were measured monthly before the animals were sacrificed. Fasting blood glucose levels were measured monthly using an Accu-Chek glucometer (Roche Diagnostics GmbH, Mannheim, Germany).

### 2.6. Echo-MRI

Echo-MRI (Whole Body Composition Analyzer, Houston, TX, USA) was performed on animals to quantify body fat mass.

### 2.7. Glucose Tolerance Test (GTT)

GTT was performed as previously described [17]. Briefly, mice were intraperitoneally injected with d-glucose (2 g/kg; Sigma-Aldrich) and blood glucose levels were measured before and after the injection using an Accu-Chek glucometer.

### 2.8. Measurement of Metabolic Parameters

After an overnight fast, all mice were intraperitoneally anesthetized with zoletil (5 mg/kg; Virbac Laboratories, Carros, France) and blood samples were collected transcardially through the left ventricle with a 1-mL syringe. Serum glucose, alanine aminotransferase (ALT), and aspartate aminotransferase (AST) levels were determined in Green Cross Reference Laboratory (Yongin-si, South Korea). Serum concentrations (*n* = 3–10) of leptin and insulin were determined using leptin (R&D Systems, MN, USA) and insulin (Shibayagi Co., Gunma, Japan) mouse enzyme-linked immunosorbent assay (ELISA) kits, according to the manufacturers’ protocols.

### 2.9. Tissue Collection and Histological Examination

For tissue analysis, mice (*n* = 3 per group) were anesthetized with zoletil and perfused with 4% paraformaldehyde (PFA) in 0.1 M phosphate-buffered saline (PBS). After 6 h of postfixation in the same fixative, the brains were removed and sequentially immersed in the sucrose solution at 15%, 20%, and finally 30% at 4 °C until they sank. The brains were then frozen and sliced into 30 μm sections. The liver and whole separating pancreas from the stomach and duodenum were processed for paraffin embedding and sliced into 5 μm sections. Pancreas and liver sections were deparaffinized and stained using hematoxylin and eosin (H&E). The stained tissue sections were visualized under a BX51 light microscope (Olympus, Tokyo, Japan). The percentage of pancreatic islet area was obtained from selected images using i-Solution (IMT i-Solution Inc., Vancouver, BC, Canada). Three fields (150 × 150 µm^2^) were randomly selected on each section from two continuous sections (*n* = 3 per group). The liver sections (*n* = 3 per group) stained by H&E-stained were examined by the histological scoring system for nonalcoholic fatty liver disease (NAFLD) activity, which was carried out by an experienced pathologist without prior knowledge of the treatment groups. The score for NAFLD activity was quantified by addition of steatosis (0–3), lobular inflammation (0–2), and hepatocellular ballooning (0–2).

### 2.10. Immunofluorescence

For double immunostaining, free-floating brain sections were incubated with mouse anti-glial fibrillary acidic protein (GFAP, 1:200, Sigma-Aldrich) and rabbit anti-ionized calcium-binding adaptor molecule-1 (IBA-1, 1:200, Wako Pure, Osaka, Japan) or rabbit anti-phospho-Drp1 (Ser 616, 1:200, Cell signaling, MA, USA) either mouse anti-neuronal nuclei (NeuN, 1:200, Millipore, MA, USA) or rabbit anti-parvalbumin (1:200, Abcam) at 4 °C overnight and washed; sections were then incubated with the donkey secondary antibodies conjugated with Alexa Fluor 488- and 594-conjugated (Invitrogen, Carlsbad, USA). The images of the sections were visualized under a BX51-DSU microscope (Olympus).

### 2.11. Duolink Immunofluorescence

To determine whether two molecules were in sufficient proximity to interact, the Duolink^®^ in situ proximity ligation assay (PLA) from the OLINK Bioscience (Uppsala, Sweden) was used, according to the manufacturer’s manuals. The anti-rabbit MINUS (first PLA probe) binds to the mouse GLP-1 antibody, whereas the anti-goat PLUS (second PLA probe) binds to the rabbit GLP-1R antibody. After performing the Duolink assay, sections were stained with DAPI (1:10,000, Thermo Fisher Scientific), washed, and a coverslip was mounted using mounting medium. A negative control experiment was performed where only GLP-1 antibody was incubated with the PLA probes. The images of sections were captured by a BX51-DSU microscope (Olympus).

### 2.12. Western Blot Analysis

For protein extraction, tissue samples of whole pancreas and hippocampi (*n* = 3–5 per group) were dissected from the mice and individually transferred to sterile 1.5-mL microcentrifuge tubes containing a lysis buffer and homogenized, as previously described [17]. To extract total hippocampi from mice, we first dissected the layer of the cerebral cortex and opened them. The exposed hippocampi were cut out from whole brain. Following the manufacturer’s instructions, the nuclear fractions were obtained from those tissues using NE-PER^®^ Nuclear and Cytoplasmic Extraction Kit (Pierce, Rockford, IL, USA). The Mitochondrial Isolation Kit (Thermo Fisher Scientific) was used to obtain the mitochondria from hippocampi. Proteins were immunoblotted with primary antibodies: GLP-1 (1:1000, Santa Cruz), GLP-1R (1:1000, Abcam), p-Tau (Ser 396, 1:1000, Thermo Fisher Scientific), total Tau (1:1000, Santa Cruz), GFAP (1:1000, Sigma-Aldrich), IBA-1 (1:1000, Wako), NF-κBp65 (1:1000, Santa Cruz), HO-1 (1:1000, StressGen, MI, USA), insulin receptor (IR)-β (1:1000, Santa Cruz), p-Drp1 (1:1000, Cell signaling), total Drp1 (1:1000, BD Bioscience), optic atrophy1 (OPA1, 1:1000, BD Bioscience), hippocalcin (1:1000, Abcam), and calbindin (1:1000, Santa Cruz). The membranes were probed with horseradish peroxidase-conjugated secondary antibodies (1:1000, Thermo Fisher Scientific) and were visualized with an enhanced chemiluminescence substrate (Pierce, Rockford, IL, USA). Nuclear p84 (1:3000, Abcam), mitochondrial VDAC1 (1:1000, Abcam), β-actin (1:5000, Sigma-Aldrich), and α-tubulin (1:5000, Sigma-Aldrich) were used as internal controls for normalize protein contents in tissue samples. The Multi-Gauge V 3.0 image analysis program (Fujifilm, Tokyo, Japan) was used for the band densitometry.

### 2.13. Morris Water Maze (MWM) Test

A MWM test was performed as previously described [17]. During the MWM test, all experimental groups and mice (*n* = 7–10 per group) were subjected to four trials per day for 4 consecutive days. The escape latency and swimming distance to find the hidden platform was recorded by a video-tracking program (Noldus EthoVision XT7, Noldus Information Technology, Netherlands). On the day of testing, the platform was removed and time spent in the target quadrant zone, where the platform had been located during training, was analyzed for 1 min.

### 2.14. Statistical Analysis

All statistical analyses were done with PRISM 7 (GraphPad Software Inc., San Diego, CA, USA). Statistical differences were determined by a one-way analysis of variance (ANOVA) followed by Turkey’s test as a post hoc analysis. For measurements of body weight and food intake, a repeated-measures analysis of variance was used. All values in plots were expressed as the mean ± standard error of the mean (SEM). The statistical significance was assessed with a *p*-value of <0.05.

## 3. Results

### 3.1. Expression, Purification, and Pharmacokinetic (PK) Evaluation of ExA

The ExA was expressed as a soluble protein from *E. coli* and successfully purified using Talon metal affinity chromatography. As shown in Figure 1B, the final ExA product after purification was clearly identified from the SDS-PAGE gel. The average yield of the ExA was 16.3 mg per liter culture, and the purity measured by densitometry analysis (image J software) was over 70%. The PK profiles of ExA were evaluated in ICR mice. Figure 1C shows the changes of ExA concentrations (μg/mL) in plasma as the function of time after subcutaneous administration of ExA in the mice. PK parameters were calculated based on the standard NCA method. As a result, the ExA exhibited a plasma half-life of 5.7 days, and the maximum plasma concentration (4.5 μg/mL on average) of ExA was reached after 2.3 h of post-administration. This plasma half-life of ExA was remarkably (57-fold) longer than the unmodified exendin-4 (*t*_1/2_: 2.4 h) [18].

### 3.2. Effects of ExA Treatment on Food Intake, Body Weight, and Blood Glucose Levels in HFD/STZ Mice

To explore the effects of ExA on food intake, body weight, and blood glucose levels, we monitored changes in HFD/STZ mice after weekly intraperitoneal injections of ExA for 6 weeks (Figure 2A). Although HFD feeding induced a reduction in food intake (g), HFD/STZ increased body weight and fasting glucose levels after 8 weeks of HFD feeding because the total calorie intake (kcal) of HFD/STZ mice was higher than ND-fed mice (Figure 2B–D and Appendix A). After 6 weeks of ExA treatment, both body weight and fat mass were significantly reduced in HFD/STZ mice (Figure 2E,F). In addition, there was a significant increase in serum leptin levels in HFD/STZ mice compared to ND-fed mice (Figure 2G). However, there was no reduction in circulating leptin levels in ND-fed mice or HFD/STZ mice after the ExA administration.

### 3.3. Effects of ExA Treatment on Insulin Resistance and Pancreatic GLP-1/GLP-1R Expression Levels in HFD/STZ Mice

Exendin-4 enhances glucose-dependent insulin secretion by pancreatic β-cells [19,20]. We thus examined the hypoglycemic effects of ExA treatment on HFD/STZ mice (Figure 3). Compared with ND-fed mice, HFD/STZ mice had higher fasting serum glucose levels, which were reduced after ExA treatment (Figure 3A). There was no significant difference between ND and ND+ExA mice (*p* = 0.089). However, there were no significant changes in serum insulin levels in any group (Figure 3B). GTT findings also demonstrated that ExA improved blood glucose regulation compared with HFD/STZ mice (Figure 3C,D). Compared with ND-fed mice, diabetic mice exhibited decreased areas of pancreatic islets (Figure 3E). Interestingly, ExA slightly reversed the reduction in islet number in diabetic mice. Finally, we determined whether ExA affected GLP-1/GLP-1R expression levels in diabetic mice (Figure 3F). Western blot analyses showed that ExA treatment promoted diabetes-induced pancreatic GLP-1 expression. In particular, treatment of ND-fed mice with ExA increased the expression of GLP-1 compared with untreated ND-fed mice. However, there were no significant changes in GLP-1R expression in either group prior to ExA treatment.

### 3.4. Effects of ExA Treatment on Hepatic Steatosis in HFD/STZ Mice

Long-term exendin-4 treatment is associated with significant improvements in abnormal hepatic enzyme levels [21]. To investigate the protective effects of ExA on hepatic steatosis in HFD/STZ mice, we measured the levels of serum hepatic enzymes, AST and ALT, and performed histological analysis for NAFLD activity (Figure 4). Increased hepatic weight in diabetic mice was significantly reversed after ExA treatment (Figure 4A,B). ExA-treated diabetic mice had significantly lower hepatic enzyme levels than untreated mice (Figure 4C,D). Histological examination revealed decreased lipid accumulation in the liver tissue of ExA-treated mice compared with untreated mice (Figure 4E). Further, ExA-treated mice exhibited reduced liver damage, as assessed by NAFLD activity scores (Figure 4F), compared with untreated mice.

### 3.5. Effects of ExA Treatment on Memory Deficits and Tau Phosphorylation in HFD/STZ Mice

In addition to its protective effects on insulin resistance and hepatic damage, some studies have demonstrated that exendin-4 enhances cognitive performance in diabetic mice [12,22]. Similarly, ExA-treated mice showed improved escape latency and fewer memory deficits compared with untreated mice, according to the results of the MWM test (Figure 5A,B). No significant differences in swimming distances were observed in either group (Figure 5C). The swimming trajectories demonstrated more directed movements to the target quadrant in ExA-treated mice compared with untreated mice, which had trajectories spread over all quadrants (Figure 5D). In addition, HFD/STZ mice had increased expression levels of p-tau compared with ND-fed mice, which were significantly reversed after ExA treatment (Figure 5E).

### 3.6. Effects of ExA Treatment on Neuroinflammation and HO-1 Expression in HFD/STZ Mice

We determined if the expression of glial activation (GFAP and IBA-1) and neuroinflammatory and oxidative stress (nuclear NF-κβ and HO-1) markers were affected by ExA treatment in diabetic mice (Figure 6). ExA treatment significantly reduced hippocampal GFAP and IBA-1 expression in diabetic mice (Figure 6A). Immunofluorescent staining also revealed the presence of large GFAP- and IBA-1-expressing glial cells in HFD/STZ mice compared with ND-fed mice (Figure 6B). However, no large GFAP- and IBA-1-positive cells were observed after ExA treatment. Furthermore, nuclear NF-κβ p65 and HO-1 expression levels were significantly increased in hippocampal sections of diabetic mice, whereas ExA treatment reduced these levels (Figure 6C,D).

### 3.7. Effects of ExA Treatment on GLP-1/GLP-1R Activation and Insulin Receptor (IR)-β Expression in HFD/STZ Mice

GLP-1 crosses the blood–brain barrier (BBB) to directly modulate neurotransmitter release and long-term potentiation [14,23]. Thus, we determined whether ExA could affect hippocampal GLP-1/GLP-1R expression levels in diabetic mice (Figure 7A). ExA treatment increased hippocampal GLP-1 expression in both ND-fed and HFD/STZ mice (Figure 7B). There were slightly reduced hippocampal GLP-1R expression levels in HFD/STZ mice compared with ND-fed mice. Although interactions between GLP-1 and GLP-1R were observed in hippocampal neurons, as assessed with the Duolink assay, and little DAPI nuclear staining was present in hippocampal neurons in HFD/STZ mice compared with the ND-fed mice, these findings were reversed following ExA treatment (Figure 7C). Few signal was observed when the GLP-1 antibody was used alone in control experiments (Figure 7D). Compared with ND-fed mice, hippocampal IR-β expression was significantly reduced in HFD/STZ mice, which was reversed following ExA treatment (Figure 7E). 

### 3.8. Effects of ExA Treatment on Hippocampal Mitochondrial Fission and Calcium-Binding Protein (CaBP) Expression in HFD/STZ Mice

Diabetes causes a significant disruption of mitochondrial structure and function, leading to increased mitochondrial fission in dorsal root ganglion neurons [24]. To investigate whether ExA treatment affects mitochondrial dynamics in the hippocampus of HFD/STZ mice, we examined mitochondrial DRP1 and OPA1 expression levels (Figure 8A–D). ExA treatment reduced DRP1 phosphorylation and numbers of p-DRP1-positive neurons in the HFD/STZ hippocampus (Figure 8A–C). However, neither HFD/STZ feeding nor ExA treatment affected the expression of mitochondrial fusion related OPA1 in the hippocampus (Figure 8D). Mitochondrial biogenesis is associated with neuronal calcium homeostasis [25,26]. Studies on age-related cognitive impairment revealed the dysregulation of CaBP-expressing interneurons under metabolic stress [27,28]. Notably, immunofluorescent staining revealed that p-DRP1 is particularly abundant in parvalbumin-positive interneurons in the hippocampus of ND-fed and HFD/STZ mice (Figure 8E). We then asked whether the decreased expression of other CaBPs in response to ExA treatment could play a role in calcium homeostasis in HFD/STZ. HFD/STZ-induced increases in hippocampal hippocalcin and calbindin protein expression levels were reversed after ExA treatment (Figure 8F,G).

## 4. Discussion

In this research, the novel exendin-4 fusion protein agent was characterized for pharmacokinetic profiles and drug efficacy in the diabetes mice models. Considering the strategy to cure metabolic and cognitive complications following diabetes, it is important to evaluate pharmacokinetic information and pharmacological activities in related tissues to ascertain the contribution of the novel fusion protein agent for the therapy. The main finding of the study was that weekly treatments of ExA for a duration of 6 weeks after the induction of diabetes improves cognitive impairment, insulin resistance, and hepatic steatosis via the activation of GLP-1/GLP-1R. These findings strongly suggest that ExA—a fusion protein—exhibits multiple therapeutic effects that help reduce the need for polypharmacy in obese and elderly diabetic patients living with additional metabolic complications.

GLP-1 is an intestinal hormone and neurotransmitter [29]. The GLP-1R agonist liraglutide suppresses appetite via proopiomelanocortin activation and the Agouti-related peptide inhibition in the arcuate nucleus of the hypothalamus [30]. In addition, the intracerebroventricular injection of exendin-4 into the fourth ventricle of the brains of rats suppressed food intake [31]. However, the specific brain regions that mediate these actions are poorly understood. In the peripheral nervous system, GLP-1R agonists, including exendin-4 and liraglutide, cause suppression of food intake via subdiaphragmatic vagal deafferentation [32]. Although the present study did not examine the impact on the brain, it is assumed that ExA administered by intraperitoneal injection may reduce food intake by acting on any hypothalamic neurons. In this study, there were no differences in serum leptin levels in ND-fed and HFD/STZ mice with or without ExA treatment. Although ob/ob mice are congenitally deficient in leptin, the exendin-4 treatment decreased food intake without affecting serum leptin levels in ob/ob mice for 10 weeks [33]. We found significant reductions in body weight and food intake after ExA treatment in diabetic mice compared with ND-fed mice. Further studies of dietary control mechanisms induced by leptin in the hypothalamus may provide a better understanding of the effects of ExA on the regulation of appetite.

GLP-1 and GLP-1R agonists, including exendin-4, stimulate insulin secretion from pancreatic β-cells only in the diabetic state; however, as a medical therapy GLP-1 is ineffective as it has a very short half-life in vivo [34,35]. Alternatively, exendin-4, a protease-resistant GLP-1R agonist, was approved for adjunct therapy to improve glycemic control in patients with T2DM who are taking metformin [36]. However, the plasma half-life (2.4 h) of exendin-4 is far from satisfying and requires twice-daily administration for sufficient glycemic control. In this regard, we developed a long-lasting exendin-4 fusion protein—ExA. ExA showed a remarkably long plasma half-life of 5.7 days in mice; allowing once-weekly administration for the treatment. Notably, we demonstrated that the partial loss of pancreatic β-cells induced by HFD/STZ was reversed after ExA treatment; furthermore, insulin secretion from ExA-treated HFD/STZ mice was well regulated in the diabetic state. Although ExA treatment particularly increased pancreatic GLP-1 expression in ND-fed mice, ExA did not increase the risk of hypoglycemia when administered to ND-fed mice.

In addition to decreasing glucose levels and stimulating weight loss, GLP-1R agonists have been used to treat T2DM accompanied by NAFLD [37,38]. GLP-1R agonist-induced weight loss reduces hepatic steatosis and increases insulin sensitivity in GLP-1R-expressing hepatocytes [39]. In accordance with several studies that showed that exendin-4 causes a significant reduction in ALT levels in diabetic patients with NAFLD [21,37], we found that ExA treatment for 6 weeks contributed to significant reductions in serum ALT and AST levels in HFD/STZ mice. Thus, our findings demonstrate that GLP-1R, as a target of ExA, may prevent or ameliorate hepatic steatosis in obese and diabetic mice. 

Because GLP-1R is expressed in lung, adipose tissue, islet cells, liver, and brain, GLP-1R agonists can elicit a variety of actions [40]. Specifically, GLP-1R agonists can cross the BBB and may serve as neuroprotective agents for Parkinson’s disease and AD [10,41,42]. In this regard, we compared GLP-1 expression in both the pancreas and hippocampus of ND-fed and ExA-treated HFD/STZ mice. As expected, both pancreatic and hippocampal GLP-1 expression levels were increased by ExA. These data indicate that systemic injection allows ExA to cross the BBB and activate hippocampal GLP-1R expression in diabetic mice.

Additionally, GLP-1 mimetics (e.g., exendin-4 or liraglutide) elicits neuroprotective effects in diabetic patients with cognitive impairment [11]. Similarly, our findings showed that ExA treatment for 6 weeks improved memory deficits in HFD/STZ mice. Therefore, GLP-1R agonists are not only neuroprotective but also have anti-inflammatory effects. Cao et al. demonstrated a reduction in glial activation by GLP-1 in a mouse model of Parkinson’s disease [43]. In lipopolysaccharide-treated rat glial cells, the activities of microglia and astrocytes, which are involved in the inflammatory response, were inhibited by GLP-1 [44]. Similar to our findings for HO-1, the antioxidant effects of exendin-4 were demonstrated by its ability to reduce oxidative stress in hyperglycemic mice [45]. Thus, the data suggests that the neuroprotective roles of ExA may be secondary to its ability to reduce inflammation and improve insulin resistance in the brain, which results from its roles in weight loss and glycemic control.

Neuronal mitochondrial fission is an alternative response to a sustained metabolic load in the diabetic brain [24]. In particular, increased mitochondrial fission is essential for hippocampal neurons to elicit changes in mitochondrial structure, neuronal function, synaptic plasticity, and cognitive function, which all place a high metabolic load on neurons [46,47]. Increased mitochondria number and rate of mitochondrial fission have been observed in dorsal root ganglia of db/db mice [24]. These data suggest that mitochondrial fission results in aberrant mitochondrial morphology. For example, we found that HFD/STZ induces increased mitochondrial DRP1 phosphorylation in the hippocampus, but no changes in the levels of the mitochondrial fusion protein, OPA1, were observed in diabetic brains. However, ExA treatment reversed hippocampal p-DRP1 expression to normal levels. These data suggest that under hyperglycemia-induced metabolic stress, mitochondrial biogenesis is normalized via GLP-1/GLP-1R activation.

In our previous study, although we did not directly measure free cytosolic Ca^2+^ levels, there was an increase in hippocalcin and S100β levels in hyperglycemic ob/ob mice, which may account for the disturbance in Ca^2+^ homeostasis [48]. Therefore, we suggest that impaired Ca^2+^ homeostasis in the brain may play a role in the neurological complications of diabetes, as treatment with a Ca^2+^ channel blocker improves Ca^2+^-dependent synaptic plasticity in the hippocampus of diabetic patients [26]. Other studies have revealed selective changes in the subtypes of GABAergic interneurons in the hippocampus of diabetic rats [49], categorized by the expression of parvalbumin, calbindin, or calretinin [50]. Larsson M. et al. recently demonstrated that the average number of parvalbumin-positive interneurons was increased in the striatum of type 2 diabetic Goto-Kakizaki rats [51]. Parvalbumin-positive interneurons are fast-spiking neurons that demand high energy for normal functioning and play an important role in cortical information processing [52]. Levels of other CaBPs, including hippocalcin and calbindin, were also increased in the hippocampus of diabetic mice, which were reduced after ExA treatment. Thus, we confirmed that Ca^2+^ dysregulation over a long period of time might lead to cognitive impairment and that increases in free cytosolic Ca^2+^ levels in the hippocampus of HFD/STZ mice could be reversed after ExA treatment.

## 5. Conclusions

In conclusion, the novel fusion protein agent ExA expanded the applications of GLP-1R agonists for improving the memory deficits in diabetic mice with its long half-life and neuroprotective activity. This preclinical study provided significant evidences of neuroprotective effects by ExA, thereby suggesting exendin-4 agents as a potential treatment against diabetes-associated cognitive decline. Based on the results, ExA had a 57-fold increased half-life, compared to unmodified exendin-4. It was found that with the long-lasting property, ExA decreased weight gain, insulin resistance, hepatic damage, and memory deficits in HFD/STZ mice. Furthermore, we demonstrated that ExA normalized mitochondrial biogenesis and CaBP-related signaling, which affect neuronal synaptic plasticity. The ExA-induced increase in insulin sensitivity and GLP-1 expression in the hippocampus may stabilize mitochondrial biogenesis and neuronal Ca^2+^ homeostasis, thus protecting against neuroinflammation and neurodegeneration in diabetic mice. Ultimately, our study proved the metabolic and neurological activities of ExA to alleviate obesity-induced diabetes and cognitive impairment, and identified long-lasting exendins as potential monotherapies for metabolic complications.

## Figures and Tables

**Figure 1 pharmaceutics-12-00159-f001:**
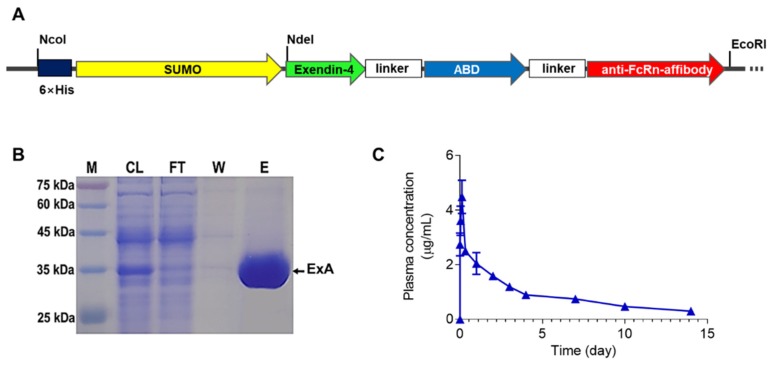
Production and pharmacokinetic (PK) characterization of long-acting exendin-4-ABD-anti-FcRn affibody (ExA). (**A**) Schematic image of the ExA gene. (**B**) SDS-PAGE results of produced ExA: M, a size marker or protein; CL, *E. coli* cell lysate; FT, flow-through solution; W, washed fraction; E, eluent buffer. (**C**) Plasma concentration-versus-time profiles of the ExA in ICR mice.

**Figure 2 pharmaceutics-12-00159-f002:**
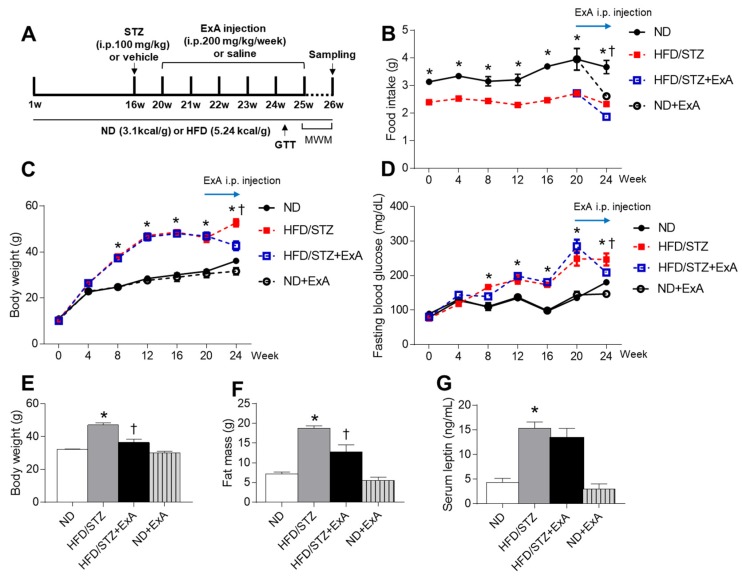
Effects of long-acting exendin-4-ABD-anti-FcRn affibody (ExA) on food intake and body weight in HFD/STZ mice. (**A**) Schematic drawing of experimental schedule. Glucose tolerance test (GTT) and Morris Water Maze (MWM) test were performed before mice sacrificed. (**B**) Food intake, (**C**) body weight, and (**D**) fasting blood glucose levels in ND-fed and HFD/STZ mice with or without ExA treatment over time. (**E**) Body weight, (**F**) fat mass, and (**G**) serum leptin levels 6 weeks after ExA treatment. Data are shown as the mean ± SEM. *: *p* < 0.05 vs. ND-fed mice, †: *p* < 0.05 vs. HFD/STZ mice.

**Figure 3 pharmaceutics-12-00159-f003:**
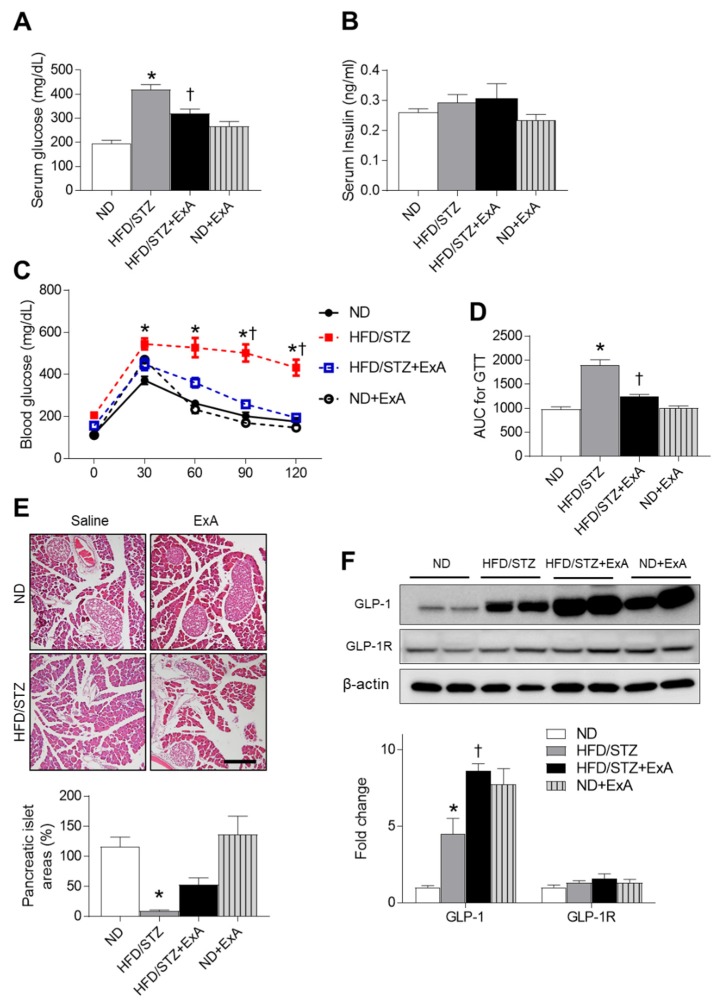
Effects of long-acting exendin-4-ABD-anti-FcRn affibody (ExA) on glucose tolerance and pancreatic GLP1 expression in HFD/STZ mice. (**A**) Serum glucose and (**B**) insulin levels in ND-fed and HFD/STZ mice with or without ExA treatment (HFD/STZ+ExA). (**C**) Glucose tolerance test (GTT) and (**D**) area under the curve (AUC) for the GTT. (**E**) Representative microscopic images of H&E-stained pancreatic sections. Scale bar = 100 µm. (**F**) Western blot and quantitative analysis results showing pancreatic GLP-1 and GLP-1R expression levels using β-actin as a loading control (*n* = 3–4 mice per group). Data are shown as the mean ± SEM. *: *p* < 0.05 vs. ND-fed mice, †: *p* < 0.05 vs. HFD/STZ mice.

**Figure 4 pharmaceutics-12-00159-f004:**
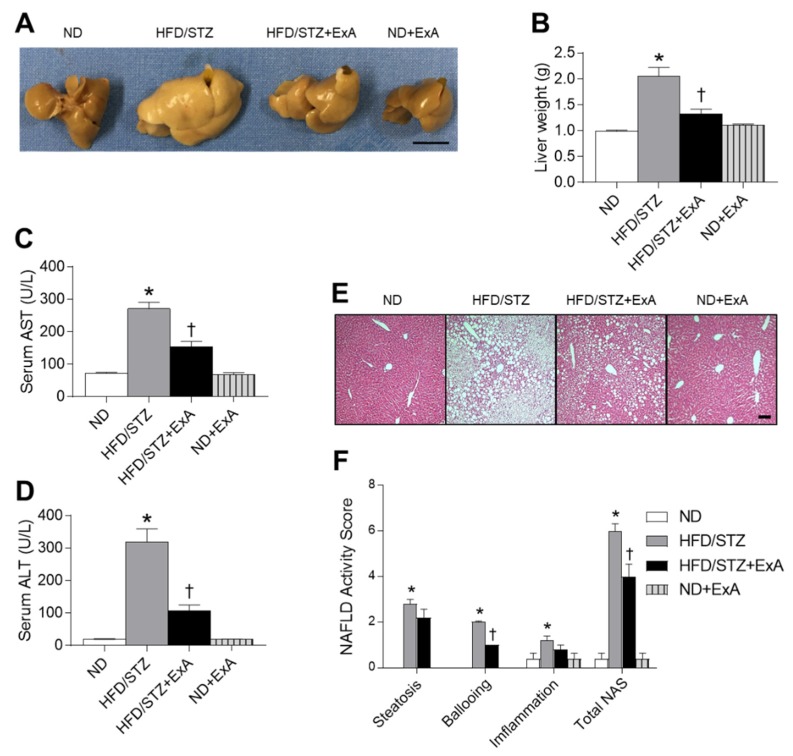
Effects of long-acting exendin-4-ABD-anti-FcRn affibody (ExA) on fatty liver in HFD/STZ mice. (**A**) Representative liver images. (**B**) Liver weight, and serum AST (**C**) and ALT (**D**) levels in ND-fed and HFD/STZ mice with or without ExA treatment. (**E**) Representative microscopic images of H&E-stained liver sections. Scale bar = 100 µm. (**F**) NAFLD activity score. Data are shown as the mean ± SEM. *: *p* < 0.05 vs. ND-fed mice, †: *p* < 0.05 vs. HFD/STZ mice.

**Figure 5 pharmaceutics-12-00159-f005:**
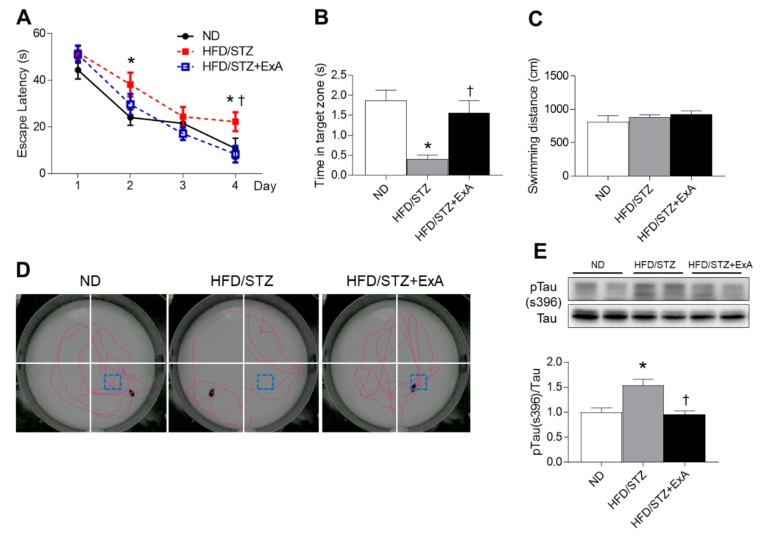
Effects of long-acting exendin-4-ABD-anti-FcRn affibody (ExA) on memory deficits and hippocampal tau phosphorylation levels in HFD/STZ mice. (**A**) Escape latency over 4 days during Morris water maze training in ND-fed mice, HFD/STZ mice, and ExA-treated HFD/STZ mice (HFD/STZ+ExA). (**B**) Average time spent in the target zone, (**C**) swimming distance, and (**D**) representative swimming paths during the probe trial. (**E**) Western blot and quantitative analysis results showing phosphorylated tau (S396) and total tau expression in the hippocampus (*n* = 3–4 mice per group). Data are shown as the mean ± SEM. *: *p* < 0.05 vs. ND-fed mice, †: *p* < 0.05 vs. HFD/STZ mice.

**Figure 6 pharmaceutics-12-00159-f006:**
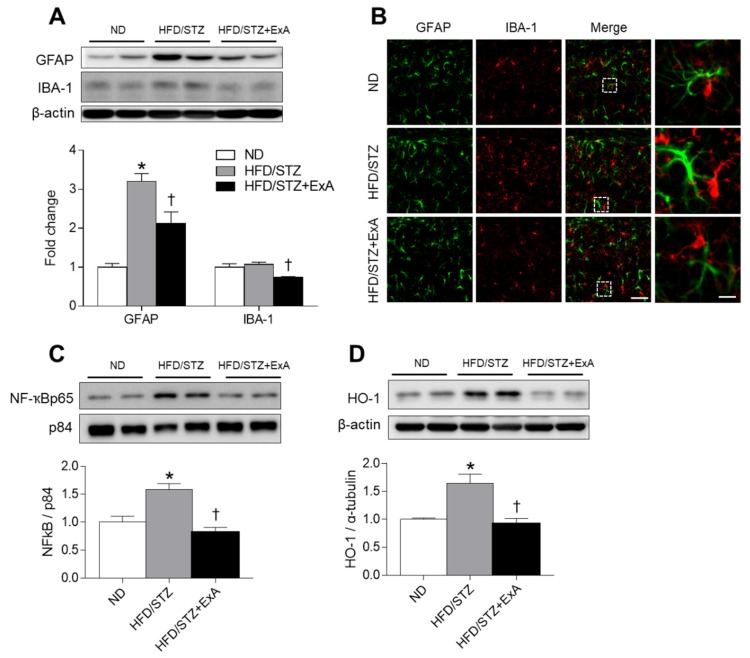
Effects of long-acting exendin-4-ABD-anti-FcRn affibody (ExA) on glial activation, NF-κβ, and HO-1 expression levels in the hippocampus of HFD/STZ mice. (**A**) Western blot and quantitative analysis results showing GFAP and IBA-1 expression in the hippocampus. (**B**) Immunofluorescence staining for GFAP (green) and IBA-1 (red) in the hippocampal CA1 regions of ND-fed mice, HFD/STZ mice, and ExA-treated HFD/STZ (HFD/STZ+ExA). Scale bar = 20 µm. (**C,D**) Western blot analysis showing NF-κβp65 (**C**) and HO-1 (**D**) expression in the hippocampus. Both p84 and β-actin were used as loading controls, respectively, for panels C and D. Data (*n* = 3–4 mice per group) are shown as the mean ± SEM. *: *p* < 0.05 vs. ND-fed mice, †: *p* < 0.05 vs. HFD/STZ mice.

**Figure 7 pharmaceutics-12-00159-f007:**
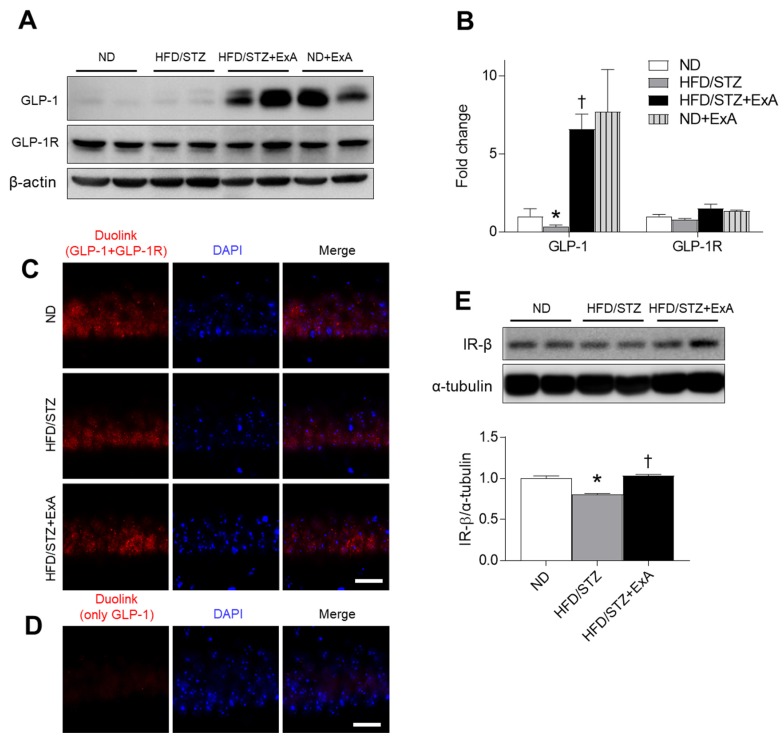
Effects of long-acting exendin-4-ABD-anti-FcRn affibody (ExA) on hippocampal GLP-1, GLP-1R, and IR-β expression levels in HFD/STZ mice. (**A,B**) Western blot analysis (**A**) and quantitative analysis (**B**) results showing GLP-1 and GLP-1R expression in the hippocampus. (**C**) Representative images of Duolink assay showing the interaction between GLP-1 and GLP-1R in the hippocampal CA1 region. Nuclei were stained with DAPI. (**D**) Negative control PLA with GLP-1 antibody only. Scale bar = 20 µm. (**E**) Western blot and quantitative analysis results showing IR-β expression in the hippocampus. Both β-actin (**A**) and α-tubulin (**E**) were used as loading controls. Data (ˆ = 3–4 mice per group) are shown as the mean ± SEM. *: *p* < 0.05 vs. ND-fed mice, †: *p* < 0.05 vs. HFD/STZ mice.

**Figure 8 pharmaceutics-12-00159-f008:**
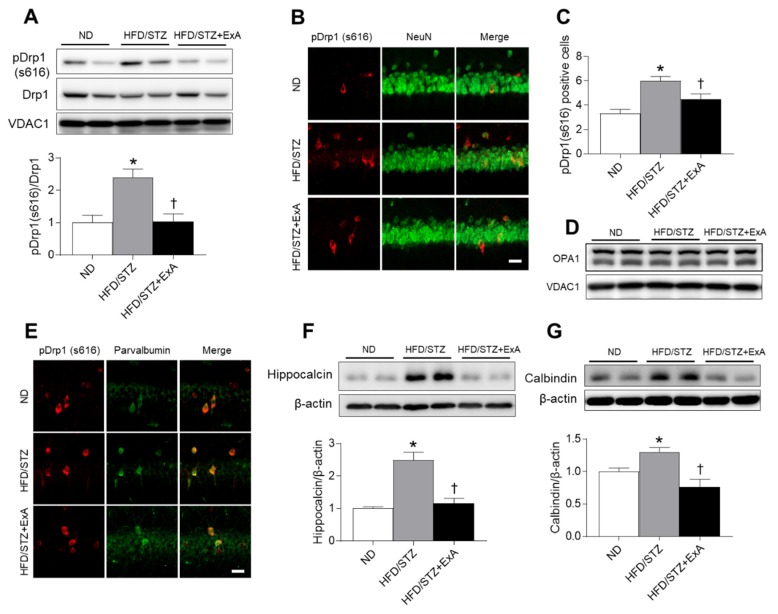
Effects of long-acting exendin-4-ABD-anti-FcRn affibody (ExA) on hippocampal DRP1 phosphorylation and calcium-binding proteins (CaBPs) in HFD/STZ mice. (**A**) Western blot and quantitative analysis results showing pDRP1 (s616) and DRP1 in the hippocampus of ND-fed and ExA-treated and untreated HFD/STZ mice. VDAC1 was used as a loading control. The ratio of pDRP1 to DRP1 was determined after normalization to VDAC1 levels. (**B**) Representative images of immunofluorescence staining for pDRP1 (red) and NeuN (green) in the hippocampal CA1 regions. (**C**) The numbers of pDRP1-positive neurons in B. (**D**) Western blot analysis results showing OPA1 protein in the hippocampus. (**E**) Immunofluorescence staining for pDRP1 (red) and parvalbumin (green) in the hippocampal CA1 regions. (**F** and **G**) Western blot and quantitative analysis results showing CaBPs (hippocalcin and calbindin) in the hippocampus. β-actin was used as a loading control. Data (*n* = 3–4 mice per group) are shown as the mean ± SEM. *: *p* < 0.05 vs. ND-fed mice, †: *p* < 0.05 vs. HFD/STZ mice.

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
