# Peer review of "Long-Lasting Exendin-4 Fusion Protein Improves Memory Deficits in High-Fat Diet/Streptozotocin-Induced Diabetic Mice"

_pharmaceutics, 2020, doi:10.3390/pharmaceutics12020159_

Round 1

Reviewer 1 Report

In this manuscript, entitled “Long-lasting exendin-4 fusion protein improves memory deficits in high-fat diet/streptozotocin-induced diabetic mice,” the authors show here effects of GLP-1 mimetic exendin-4 fusion protein on memory loss in high-fat diet/streptozotocin mouse model of type 2 diabetic mellitus.

The primary strengths of this manuscript are its careful evaluation of the pharmacokinetic profile of Ex A, its effects on insulin resistance, and markers associated with diabetic mellitus. In general, the manuscript is well-written; however, this manuscript has several critical issues that should be addressed…

Line50> It should be long-lasting. Fig1B> what are CL, FT, W and E? in the interest of readers, the authors need to write clearly in the figure legend. Line 229> “Fig 1B-D” it should be Fig 2B-D. Inconsistent writing of exendin-4-ABD-anti-FcRn affibody in figure legends. Fig3A>Please explain why there is an increase in serum glucose level in ND+ExA group vs ND? Fig3F> I assume that Western blot was run with two different samples. In ND+ExA group one of GLP-1 bands is normal whereas another band is highly induced after ExA treatment, why? Line302> it should be “markers”. Fig6C&D> I am not convinced with the data in Fig6 C&D, it does not look to me a significant reduction in NF-kBp65 and HO-1 expression after ExA treatment of HFD/STZ group. They need to provide better Western blot data. Fig7C> To rule out non-specific signals, the authors need to show no-antibody and single antibody stained figures.

Author Response

Response to Reviewer 1’ comments

Line50> It should be long-lasting.

We corrected it

Fig1B> what are CL, FT, W and E? in the interest of readers, the authors need to write clearly in the figure legend.

We revised the figure legend of Figure 1 by adding the details of the following abbreviations: CL, FT, W, and E. Briefly, CL indicates cell lysate, FT is the flow-through, W is wash fraction, and E is the eluent from the Talon resin purification.

Line 229> “Fig 1B-D” it should be Fig 2B-D. Inconsistent writing of exendin-4-ABD-anti-FcRn affibody in figure legends.

We corrected the typo (from Fig. 1B-D to Fig. 2B-D) and rectified the inconsistent writing of the name of “exendin-4-ABD-anti-FcRn affibody” in all figure legends and text throughout the manuscript.

Fig3A>Please explain why there is an increase in serum glucose level in ND+ExA group vs ND?

Although a slight increase was observed in the serum glucose level in ND+ExA-treated mice vs ND-fed mice, there was no statistical difference (P= 0.089 by 1-way ANOVA).

Fig3F> I assume that Western blot was run with two different samples. In ND+ExA group one of GLP-1 bands is normal whereas another band is highly induced after ExA treatment, why?

We thoroughly appreciate the reviewer’s valuable comment. As we found that there was a significant difference between lane 1 and lane 2 of β-actin in ND+ExA mice, along with the GLP-1 bands’ difference in the figure 3F, the protein quantification and western blot analysis were performed again. We replaced the figure 3F in the revised manuscript. 

Line302> it should be “markers”.

We corrected this typo.

Fig6C&D> I am not convinced with the data in Fig6 C&D, it does not look to me a significant reduction in NF-kBp65 and HO-1 expression after ExA treatment of HFD/STZ group. They need to provide better Western blot data.

As the reviewer suggested, we again performed Western blot for NF-kBp65 and HO-1 and replaced them in the revised manuscript.

Fig7C> To rule out non-specific signals, the authors need to show no-antibody and single antibody stained figures. 

We appreciate the reviewer’s thoughtful comment. As the reviewer suggested, we did again performed Duolink immunofluorescence with GLP-1 antibody only. The additional figures were added to the figure 7D.

Reviewer 2 Report

section 2.9 specify how the pancreas was harvested, whole or sectioned? section 2.12 how the hippocampus portion was taken ? How was the sample size chosen for each analysis? Are the data shown representative of how much experimental replicates? in figure 2 B the ND + Exa group shows a significant reduction in food intake, and in panels C the NFD \ STZ group shows an increase in body weight in contrast to the food consumed. how do the authors explain this data? should be discussed in the text figure 3 E how were islet areas measured? figure 6,7, 8 loading controls for western blot as they were chosen? the authors might consider showing the entire developments used for membranes in the western blot as supplementary matheriarls

Author Response

Response to Reviewer 2’ comments

Section 2.9 specify how the pancreas was harvested, whole or sectioned?

The whole separating pancreas from the stomach and duodenum were used in this study. This detailed explanation was added to section 2.9 in the revised manuscript.

2, Section 2.12 how the hippocampus portion was taken ?

To extract total hippocampi from mice, we did firstly dissected the layer of cerebral cortexâ‘  and opened them. Exposed hippocampusâ‘¡ was cut out and then added into liquid nitrogen. This explanation was added to section 2.12 in the revised manuscript.

How was the sample size chosen for each analysis? Are the data shown representative of how much experimental replicates?

For each analysis, the number of experimental mice was described in materials sections. In particular, in western blot analysis, the mean values of each protein were obtained from two separate experiments (n = 3-4 mice per group).

In figure 2 B the ND + Exa group shows a significant reduction in food intake, and in panels C the HFD\STZ group shows an increase in body weight in contrast to the food consumed. how do the authors explain this data?

Because of anorexigenic effect of HFD feeding, food intake (g) is lower in HFD-fed rodents than in normal chow-fed controls. In accordance with the evidence that leptin inhibits food intake and stimulates energy expenditure in mice, HFD-fed mice have the reduced food intake, whereas the body weight of these mice is increased by high total calorie intake. So, we calculated the total calorie intake (kcal) out of the amount of food (g) consumed by ND, HFD/STZ, HFD/STZ+ExA, and ND+ExA mice. HFD contains 5.2 kcal/g (60 kcal% fat, D12492; Research Diets), and ND contains 3.1 kcal/g (2018S; Harlan Laboratories). Total calorie intake (kcal) was incorporated into the supplementary figure 1. 

Should be discussed in the text figure 3 E how were islet areas measured?

This detailed explanation was added to section 2.9 in the revised manuscript, as follwing: The percentage of pancreatic islet area was obtained from selected images using i-Solution (IMT i-Solution Inc., Vancouver, BC, Canada). Three fields (150 x 150 µm2) were randomly selected on each section from two continous sections (n = 3 per group).  

Figure 6,7, 8 loading controls for western blot as they were chosen?

As described in section 2.12 “Western blot analysis”, we used α-tubulin and β–actin as loading controls for the proteins from total lysates. Dependent on the molecular weight of the analyzed proteins, either α-tubulin (molecular weight, 50 kDa) or β–actin (molecular weight, 42 kDa) was chosen. The p84 was used as the internal control for nuclear proteins, and the VDAC1 was for mitochondrial proteins.

The authors might consider showing the entire developments used for membranes in the western blot as supplementary materials

As the reviewer commented, all membrane blot images were incorporated into the supplementary figure 2. 

Round 2

Reviewer 1 Report

The authors have addressed all of my concerns, and I do not have any further questions.

Reviewer 2 Report

the authors have sufficiently increased the quality of the manuscript